# Antibacterial Activity of Propolis-Embedded Zeolite Nanocomposites for Implant Application

**DOI:** 10.3390/ma14051193

**Published:** 2021-03-03

**Authors:** Jun Sik Son, Eun Ju Hwang, Lee Seong Kwon, Yong-Gook Ahn, Byung-Kwon Moon, Jin Kim, Douk Hoon Kim, Su Gwan Kim, Sook-Young Lee

**Affiliations:** 1IT-Bio Material Research Team, Korea Textile Development Institute, Daegu 41842, Korea; sonjk1@empas.com; 2Smart Medical Convergence Technology Support Center, Chosun University, Gwangju 61012, Korea; ejlife0827@naver.com (E.J.H.); ygookahn@naver.com (Y.-G.A.); mbkbusy@gmail.com (B.-K.M.); 3RAPHA BIO Co. Ltd., Wanju-Gun, Jeollabuk-do 55367, Korea; eskwon1946@hanmail.net; 4Dental Healthcare & Clinical Trial Center, Chosun University, Gwangju 61452, Korea; cream4251@hanmail.net; 5Research Center, Medical Division, Nexturn Co. Ltd., Gyeonggi-do 17086, Korea; doukhoon@naver.com; 6Sangmu Su Dental Clinic, Gwangju 61998, Korea; 7Regional Innovation Center for Dental Science & Engineering, Chosun University, Gwangju 61452, Korea

**Keywords:** propolis, zeolites, nanocomposites, biocompatible polymer, dental implant

## Abstract

This study investigates the potential of propolis-embedded zeolite nanocomposites for dental implant application. Propolis-embedded zeolite nanocomposites were fabricated by complexation of propolis and zeolites. Then, they were pelleted with Poly(L-lactide) (PLA)/poly(ε-caprolactone) (PCL) polymer for the fabrication of a dental implant. The chemical properties of propolis were not changed during the fabrication of propolis-embedded zeolite nanocomposites in attenuated total reflection-fourier transform infra-red (ATR FT-IR) spectroscopy measurements. Propolis was continuously released from propolis-embedded zeolite nanocomposites over one month. PLA/PCL pellets containing propolis-embedded zeolite nanocomposites showed longer sustained release behavior compared to propolis-embedded zeolite nanocomposites. Propolis-embedded zeolite nanocomposite powder showed similar antibacterial activity against *C. albicans* in an agar plate and formed an inhibition zone as well as chlorohexidine (CHX) powder. Eluted propolis solution from PLA/PCL pellets also maintained antibacterial activity as well as CHX solution. Furthermore, eluted propolis solution from PLA/PCL pellets showed significant antibacterial efficacy against *C. albicans, S. mutans* and *S. sobrinus*. Dental implants fabricated from PLA/PCl polymer and propolis-embedded zeolite nanocomposites also have antibacterial efficacy and negligible cytotoxicity against normal cells. We suggest that PLA/PCl pellets containing propolis-embedded zeolite nanocomposites are promising candidates for dental implants.

## 1. Introduction

Various nanocomposites, including polymer nanoparticles, graphene, zeolites, nanofibers, ceramic materials and metal nanoparticles, have been introduced for application in the biomedical field due to their unique features [1,2,3,4,5]. Due to their nanodimensional scales, nanocomposites have biomimetic properties and cover a wide range of applications in the field of bioengineering and regenerative medicine [1,2,3,4,5,6]. Especially, nanocomposites or nanomaterials for the modification of bioimplant surfaces have been extensively investigated to confer a longer antibacterial effect and improved biocompatibilities, since infection problems and/or biofilm formation onto implant surfaces induce functional failure and limit usage as a medical device [7,8,9,10,11]. For example, an orthodontic mini-implant coated with chitosan and antibacterial agents effectively inhibited the formation of biofilm by Porphyromonas gingivalis [8]. Nanocomposites such as Zno nanorods or nanospheres showed excellent antibacterial activities against *Escherichia coli* and *Staphylococcus aureus* [9]. Surface modification with antibacterial peptide, such as GL13K, provides the sustained inhibition of bacterial biofilm formation onto the implant surfaces [10]. Furthermore, nanocomposites and/or nanomaterials for the modification of implant surfaces can be considered as an excellent candidate to inhibit metallosis, which causes bone loss around dental implants and systemic cytotoxicity [11,12,13].

Among various nanocomposites, zeolite is a molecular sieve with a porous structure, which was formed from crystalline metal atoms, such as silicon, phosphorous, aluminum and oxygen [5]. Since zeolites are materials defined as biocompatible, edible, ion exchangeability and water absorbent materials, they have been extensively investigated in the biomedical field [6,14,15,16,17]. Kihara et al. reported that the cytotoxicity of zeolites is dependent on size, morphology and chemical composition, while pure silica-nanozeolite is non-toxic to HeLa cells [15]. Furthermore, zeolitic imidazolate frameworks (ZIF) can be used for cancer drug targeting, antibacterial treatment for wound infection and biomineralization [16]. For example, ZIFs have frequently been considered as a vehicle for an anticancer drug or diagnostic imaging agent because they have a huge surface area, porosity and functionality [18,19]. Bimetallic ZIF (Mn-ZIF-8) incorporating 5-FU has superiority in drug delivery and diagnosis using magnetic resonance imaging (MRI) [19].

Among the major causes of dental implant failure is bacterial infection and long-term usage of bisphosphonate [20,21,22]. Even though the oral administration of antibiotics was effective to reduce dental implant failure, an extremely higher dose of antibiotics is required to treat bacterial infection- or biofilm formation-related infections, and these treatment options are difficult to carry out [22,23]; that is, the higher dose of antibiotics required to achieve the minimum inhibitory concentration (MIC) may induce serious adverse effects on the human body [24,25]. To solve these problems, various drug delivery protocols, including muco-adhesive polymers, sustained release devices, liposomes and nanoparticles, have been suggested [25,26,27,28]. Strategies for antibacterial drug delivery against dental implant application still need to be developed. Since zeolite is also believed to be an excellent antiseptic material, the antibacterial activity of zeolites and their use as a platform for antibacterial agents have also been investigated by several scientists [16,17,29,30,31]. For example, silver nanoparticle-doped ZIF nanocomposites have superior antibacterial activities against Gram-positive and Gram-negative bacteria [29]. Zeolites are considered to be a good candidate for silver ions and antibacterial nanoplatforms [30]. Natural zeolites are effective carriers of silver ions and offer several advantages [31]. Song et al. reported that stimuli-sensitive ZIF nanocomposites controlled antibiotic release by light and/or pH and then efficiently inhibited the growth of bacteria [32].

In this study, we fabricated propolis-embedded zeolite nanocomposites for an antibacterial strategy of dental implant. Propolis was used as an antibacterial agent for dental implant, since it has antibacterial activity against various microorganisms with low adverse effects against normal tissues and cells [33,34,35]. Propolis-embedded zeolite nanocomposites were further complexed with biocompatible polymers to fabricate a dental implant. The antibacterial activity and cell cytotoxicity of propolis-embedded zeolite nanocomposites and zeolites/polymer composites were further complexed with polymers to fabricate a dental implant.

## 2. Materials and Methods

### 2.1. Materials

Zeolite (Zeolite H-β powder) was purchased from Zeolyst International Co., (Conshohocken, PA, USA). Propolis extracts were purchased from Rapha Propolis Co., (Jeonju, Korea). Chlorhexidine (CHX), agar, Luria Bertani broth (LB broth) and thiazolyl blue tetrazolium bromide (MTT) were purchased from Sigma-Aldrich Chem. Co. (St. Louis, MO, USA). PLA and PCL polymers were purchased from Sigma-Aldrich Chem. Co. (St. Louis, MO, USA). All chemicals and organic solvent were used as extra pure grade.

### 2.2. Fabrication of Propolis-Embedded Zeolite Nanocomposite

Empty zeolite nanocomposites: zeolite (H-beta, SiO_2_:Al_2_O_3_ = 98:2 (*w/w*)) was pre-treated at 600 °C for 1 h to remove moisture from the pore of the zeolite. Then, these were used to embed bioactive agents.

To embed propolis, propolis solution (10 g/150 mL water) was mixed with zeolite nanocomposites (20 g) and then magnetically stirred for 24 h. Following this, propolis-embedded zeolite nanocomposites were harvested by centrifugation at 15,000× *g* for 30 min, washed with deionized water and then harvested again by centrifugation. Resulting solutions were lyophilized for 2 days to obtain propolis-embedded zeolite nanocomposites.

To evaluate the loading efficiency of propolis, the concentration of remaining propolis aqueous solution and deionized water was measured using high-performance liquid chromatography (DIONEX Ultimate 3000 Standard Systems, Thermoscientific Co., Waltham, MA, USA), and then loading efficiency was calculated with the following equation: loading efficiency (%, *w/w*) = [(total amount of propolis—amount of remained propolis)/total amount of propolis] × 100.

HPLC was equipped with an Ultimate 3000 pump (Dionex Softron GmbH, Germering, Germany), Ultimate 3000 autosampler column compartment and Ultimate 3000 Variable Wavelength Detector (Dionex Softron GmbH, Germering, Germany). HPLC measurements were as follows: The mobile phase was mixtures of water/methanol (70/30 (*v/v*)) and acetic acid (0.1% (*v/v*)). The flow rate was 1 mL/min, and the injection volume was 10 μL. The quantity of propolis was measured at 275 nm using an Ultimate 3000 Variable Wavelength Detector.

For comparison, CHX-embedded zeolite nanocomposites were prepared as follows: CHX (30 mg) in water was mixed with 30 mg of zeolites, magnetically stirred for 24 h, then harvested by centrifugation. These were washed with deionized water and harvested by centrifugation followed by lyophilization for 2 days.

### 2.3. Characterization of Propolis-Embedded Zeolite Nanocomposites

Chemical properties of propolis and propolis-embedded zeolite nanocomposites were measured with Fourier transform-infrared spectra (FT-IR) (Spectrum Two, PerkinElmer, Buckinghamshire, UK). The morphology of empty zeolite nanoparticles and propolis-embedded zeolite nanocomposites was observed with a field emission scanning electron microscope (FE-SEM) (Gemini 500, Carl Zeiss, Jena, Germany). The observation of nanocomposite morphology was performed at 50 kV.

### 2.4. Preparation of PLA/PCL Pellets and Dental Implants

PLA/PCL pellets containing propolis-embedded nanocomposites were prepared as follows: 10 g of PLA/PCL (97/3, *w/w*) was melted by heating and then mixed with 1 g of propolis-embedded zeolite nanocomposites. These were stirred for 3 h and, then pellets were extruded using a twin screw extruder (Cowin extrusion, Nanjing, China) (Appendix A). Pellets were used for a drug release study or antibacterial test. Empty pellets were also prepared without addition of propolis-embedded nanocomposites.

Dental implants were prepared as follows: 10 g of PLA/PCL (97/3, *w/w*) was mixed with 1 g of propolis-embedded zeolite nanocomposites and then filled into the mould die. Then, these were packed for 6 h and then cooled for 30–40 s at room temperature for plasticization. These were ejected carefully to make a dental implant and used to study antibacterial activity.

### 2.5. Antibacterial Activity Study

Fresh colonies of various bacteria, such as *Candida albicans* (*C. albicans*)*, Streptococcus mutans* (*S. mutans*) and *Streptococcus sobrinus* (*S. sobrinus*), were obtained from Korean Collection for Type Cultures (KCTC, Jeongeup, Korea). *Escherichia coli* (*E. coli ATCC^®^ 25922^TM^*) was obtained from the American type culture collection (ATCC, Manassas, VA, USA). Each bacterium was inoculated into 10 mL of liquid Luria Bertani broth (LB broth; Biosesang, Seoul Korea) and then incubated at 37 °C for 24 h. These were inoculated onto the agar plate or culture medium (LB broth) for the antibacterial activity test.

For the antibacterial test, propolis was eluted from propolis-embedded zeolite nanocomposites and PLA/PCL pellets using ethanol for 24 h. This solution was used to measure drug concentration and diluted with bacterial culture medium more than 1000 times.

### 2.6. Cell Culture

Human epithelial keratinocyte (HaKaT) cells were obtained from the American Type Culture Collection (ATCC, Manassas, VA, USA). HaKaT cells were cultured with Dulbecco’s Modified Eagle Medium (DMEM) supplemented with 10% (*v/v*) fetal bovine serum (FBS)/1% (*v/v*) antibiotics at 37 °C in a 5% CO_2_ incubator.

To study intrinsic cytotoxicity of the dental implant against normal cells, HaKaT cells (1 × 10^4^ cell/well) in a 96-well plate were exposed to various compounds (100 μg/mL), such as media only, empty PLA/PCL pellets and PLA/PCL pellets containing propolis-embedded zeolite nanocomposites. Viability of cells was evaluated with MTT cell cytotoxicity assay. Cells were incubated with various compounds for 1 day. Following this, MTT (30 μL, 5 mg/mL in PBS) was added to the 96-well plates and then further incubated for 3 h to form formazan crystals. Then, cells were solubilized with DMSO, and the absorbance (560 nm test/630 nm reference) was determined with an Infinite M200 Pro microplate reader (Molecular Device Co., Sunnyvale, CA, USA). Results were calculated from eight wells and expressed as mean ± standard deviation (S.D.).

## 3. Results

### 3.1. Fabrication and Characterization of Zeolite Nanocomposites

As shown in Scheme 1, propolis-embedded zeolite nanocomposites were fabricated by the mixing of dehydrated zeolites and propolis aqueous solution. Propolis was selected as an antibacterial agent because it has excellent antibacterial activity against oral bacteria, such as *C. albicans*, *S. mutans* and *S. sobrinus*, as shown in Appendix A. The loading efficiency of propolis into zeolite nanocomposites was calculated as 78.6% (*w/w*) from HPLC measurement (Appendix A). Propolis was efficiently absorbed into the zeolite nanocomposites.

The results of the ATR FT-IR spectra measurements are shown in Figure 1. As shown in Figure 1, intrinsic peaks of propolis itself were observed between 700 and 3400 cm^−1^. When propolis was embedded into zeolite nanocomposites, its intrinsic peaks were also observed from the measurements of propolis-embedded nanocomposites; that is, intrinsic peaks of propolis were found as follows: 2920 cm^−1^, C-H bands of aromatic compounds; 1600 cm^−1^, C=C stretching of aromatic ring; 1200–1400 cm^−1^, C–O–C stretching, C–H wagging and CH+OH bedding; 861 cm^−1^, vibration of aromatic ring. These results indicated that the intrinsic properties of propolis were properly maintained during the drug loading process.

Figure 2 shows the morphological properties and particle size of empty (Figure 2a) and propolis-embedded zeolite nanocomposites (Figure 2b). As shown in Figure 2, propolis-embedded zeolite nanocomposites were had a brown color (Figure 2b), while empty zeolite nanocomposites revealed a white color (Figure 2a), as shown in Figure 2a. The microscopic morphology of propolis-embedded zeolite nanocomposites was not significantly changed compared to empty zeolite nanocomposites, even though they slightly aggregated each other.

### 3.2. Fabrication of PLA/PCL Pellets Complexed with Propolis-Embedded Nanocomposites and Drug Release

For the dental implant application of propolis-embedded nanocomposites, propolis-embedded nanocomposites were complexed with PLA/PCL polymers as shown in Figure 3a,b, and then pellets were fabricated using a twin screw extruder (Appendix A). Empty pellets (Figure 3a) and propolis-embedded nanocomposites—complexed pellets (Figure 3b)—were fabricated for comparison. To characterize propolis in zeolite nanocomposites and PLA/PCL pellets, a drug release study was performed in vitro as shown in Figure 3c,d. As shown in Figure 3c, propolis was continuously released from zeolite nanocomposites over one month. The burst release behavior of propolis was observed for 7 days, and then it was released in a sustained manner as shown in Figure 3c. Interestingly, the release rate of propolis became slower than that of zeolite nanocomposites when propolis-embedded nanocomposites were pelleted with PLA/PCL polymers as shown in Figure 3d. Furthermore, burst release behavior was not observed from PLA/PCL pellets, and the cumulative released fraction of propolis was less than 50% (*w/w*) for one month, indicating that the PLA/PCL pellets complexed with propolis-embedded nanocomposites prevent the burst release effect and induce the sustained release of propolis.

### 3.3. Antibacterial Activity

To evaluate the antibacterial activity of propolis-embedded nanocomposites and PLA/PCL pellets, the power of the drug and nanocomposites was used as shown in Figure 4a (Powder). Chlorohexidine (CHX) powder was used for comparison. As shown in Figure 4(a1), CHX powder clearly built an inhibition zone. Propolis-embedded zeolite nanocomposites also formed a similar inhibition zone compared to CHX powder, while empty zeolite power did not form an inhibition zone, indicating that zeolite itself has weak or no antibacterial activity (Figure 4(a2,a3)). To evaluate the changes in antibacterial activity during the pellet fabrication process, liberated propolis aqueous solution from PLA/PCL pellets was also tested using *C. albicans* as shown in Figure 4b (cotton swab). As shown in Figure 4(b1), the CHX solution in the cotton swab clearly built an inhibition zone. Liberated propolis aqueous solution from PLA/PCL pellets also formed an inhibition zone, even though its inhibition zone was smaller than that of CHX solution (Figure 4(b2)). Eluted solution from empty zeolite nanocomposites did not form an inhibition zone as expected (Figure 4(b3)). Table 1 summarizes the comparison of the inhibition zones estimated from Figure 4.

The antibacterial activity of propolis embedded in zeolite nanocomposites and PLA/PCL pellets was evaluated with various oral bacteria, such as *C. albicans, S. mutans* and *S. sobrinus*, as shown in Figure 5 and Table 2. Propolis was eluted from PLA/PCL pellets using EtOH and diluted 1000 times with aqueous solution. As shown in Figure 5, less than 10 μg/mL of eluted propolis did not clearly form an inhibition zone. An inhibition zone was formed at higher than 50 μg/mL propolis, and its area was dose-dependently increased in all bacterial strains, as summarized in Table 1. These results indicated that the fabrication process of propolis-embedded zeolite nanocomposites and PLA/PCL pellets did not significantly affect the intrinsic property of propolis and maintained antibacterial activity.

### 3.4. Biological Activity of Dental Implant Fabricated from Propolis-Embedded Zeolite Nanocomposite and PLA/PCL Polymer

Figure 6a shows the fabrication process of the dental implant using a propolis-embedded zeolite nanocomposite and PLA/PCL polymer. To make a real dental implant, a propolis-embedded zeolite nanocomposite and PLA/PCL polymer were manufactured through material filling, closed mold die, packing, cooling, open mold die and ejecting processes (Figure 6a). Then, various shapes of dental implant were obtained as shown in Figure 6b. Furthermore, an empty dental implant was also fabricated using PLA/PCL polymer in the absence of propolis-embedded zeolite nanocomposites for comparison. Figure 6c,d shows the antibacterial activity of the real dental implant fabricated in Figure 6b using *E. coli*. As shown in Figure 6c, the empty dental implant did not inhibit the growth of bacteria, and the bacterial colony was filled in the agar plate. However, the PLA/PCL dental implant containing propolis-embedded zeolite nanocomposites properly inhibited the growth of bacteria and formed significantly lower numbers of colonies in the agar plates, as shown in Figure 6d. These results indicated that the PLA/PCL dental implant containing propolis-embedded zeolite nanocomposites has antibacterial efficacy and efficiently inhibited the growth of bacteria.

Figure 7 shows the cytotoxicity of the PLA/PCL dental implant containing propolis-embedded zeolite nanocomposites against HaCaT cells. Propolis itself has no significant toxicity against HaKaT cells, i.e., viability of cells was higher than 80% at 10 μg/mL for 2 days (Appendix A). As shown in Figure 7, the PLA/PCL dental implant containing propolis-embedded zeolite nanocomposites did not inhibit the viability of normal cells, and they still had biocompatible properties.

## 4. Discussion

Zeolite nanoparticles or nanocomposites have been investigated by various investigators for medical applications [6,14,15,16,17]. Since they have no cytotoxicity against normal cells and tissues, they have been considered as an ideal candidate for drug delivery vehicles [14,15,16,36]. For example, Cao et al. reported that the tandem post-synthetic modification of ZIF containing 5-fluorouracil has potential for high drug loading, stimuli-sensitive drug release in the tumor microenvironment and synergistic anticancer effects against esophageal squamous cell carcinoma [37]. Chen et al. reported that ammonium methylbenzene blue-incorporated ZIF derivatives have superior antibacterial capacity against *E. coli, S. aureus* and *methicillin-resistant S. aureus* [38]. Specifically, ZIF-8 incorporating ceftazidime has been suggested as a suitable platform for long-term antimicrobial therapy due to its sustained release properties [39]. Furthermore, ZIF-8 incorporating ceftazidime has compatibility against lung epithelial cell lines, and these can be a useful platform for the targeting of pulmonary and/or intracellular infections, since they can be delivered intracellularly against infected normal cells [39]. Additionally, zeolites can be used to prevent the release of toxic metal ions into human tissues and improve osteointegration in the bone, since metallosis is considered as an important reason for the failure of an implant and also induces systemic cytotoxicity [12,13,40].

Propolis has been proposed as an antibacterial material for dental and oral healthcare, because it is a non-toxic material for human use [41]. Propolis-based mouthwashes have a reasonable antibacterial activity [42,43]. However, propolis has limitations in the eradication of biofilms because its antibacterial efficacy and toxicity against normal cells are dependent on the metal ions found in propolis [44]. Propolis solution has an appropriate bactericidal effect on *S. mutans* and *C. albicans*, but it has practically negligible efficacy in the eradication of denture biofilms [45]. Afrasiabi et al. reported that propolis nanoparticles provide a synergistic effect on antimicrobial photodynamic therapy against *Streptococcus mutans* [46]. The cellulose membrane incorporated with propolis-containing self-microemulsifying formulation has been shown to sustain release properties for up to 7 days and promotes would healing activity with excellent antibacterial activity [47]. Chitosan vanish-containing propolis has been shown to sustain release properties over one week and shows an improved bactericidal effect compared to CHX [48]. In our study, propolis-embedded zeolite nanocomposites showed sustained release properties over one month, as shown in Figure 3c,d. It seems that the drug release capacity of our nanocomposites was almost similar to that of other results [47]. Marquele-Oliveira et al. reported that the drug release from the biocellulose membrane continued over one week [47]. Furthermore, Franca et al. also reported that propolis was continuously released from chitosan vanish over two months [48]. Our PLA/PCL pellets incorporated with propolis-embedded zeolite nanocomposites also showed extended drug release behavior over one month, indicating that propolis-embedded zeolite nanocomposites and their polymer pellets are suitable devices for the long-term inhibition of dental/oral pathogens in implants. Furthermore, PLA/PCL pellets complexed with propolis-embedded zeolite nanocomposites showed a longer sustained release behavior compared to zeolite nanocomposites. Our dental implant platforms are also a long-term treatment for antibacterial strategies. The antibacterial activity of a real dental implant was evaluated with *E. coli ATCC 25922 strain*, because this strain is susceptible to all antibiotics and due to the ease of comparison for the antibacterial test [49]. We proved that the dental implant fabricated from PLA/PCL polymer and propolis-embedded zeolite nanocomposites has evident antibacterial efficacy against E. coli (Figure 6c,d). Furthermore, these platforms have no significant cell cytotoxicity against HaKat cells (Figure 7). Our results were quite similar to those of other investigators. Since bacterial infection and/or biofilm formation on the dental implant is closely associated with dental disease and failure of dental implant, long-term treatment strategies using sustained release implant modules are considered as ideal candidates for the success and maintenance of dental implants.

Propolis and/or its extracts have antitumor, antibacterial and anti-inflammatory effects in biological systems [50,51,52]. Specifically, the application of propolis in mouthwash is known to suppress dysphagia or mucositis efficiently in clinical trials [42,43]. Its long-term antibacterial and anti-inflammatory effects may help the settlement of the dental implant. Furthermore, it was reported that propolis nanoparticles have high efficacy in suppressing *S. mutans*-derived biofilm formation with negligible cytotoxicity against HGF-1 human gingival fibroblast cells [46]. Our results indicated that the dental implant fabricated from PLA/PCL polymer and propolis-embedded zeolite nanocomposites maintained antibacterial activity during the pellet fabrication process and, furthermore, has no significant cytotoxicity against normal cells (Figure 6 and Figure 7).

## 5. Conclusions

This is the first report on the fabrication of propolis-embedded zeolite nanocomposites and their application in dental implants for antibacterial strategies. The chemical properties of propolis were maintained during the fabrication process of propolis-embedded zeolite nanocomposites. Propolis was continuously released from propolis-embedded zeolite nanocomposites over one month, and PLA/PCl pellets containing propolis-embedded zeolite nanocomposites showed longer sustained release behavior compared to propolis-embedded zeolite nanocomposites. Propolis-embedded zeolite nanocomposite powder showed similar antibacterial activity against *C. albicans* in an agar plate and formed an inhibition zone as well as CHX powder. Eluted propolis solution from PLA/PCl pellets also maintained antibacterial activity as well as CHX solution. Eluted propolis solution from PLA/PCl pellets showed significant antibacterial efficacy against *C. albicans, S. mutans* and *S. sobrinus*. Dental implants fabricated from PLA/PCl polymer with propolis-embedded zeolite nanocomposites also have antibacterial efficacy and negligible intrinsic toxicity against normal cells. Propolis-embedded zeolite nanocomposites can be applied not only as a dental material but also as an orthopedic material due to their excellent antibacterial activity and biocompatibility.

## Data Availability

The data presented in this study are available in Son, J.S.; Hwang, E.J.; Kwon, L.S.; Anh, Y.G.; Moon, B.K.; Kim, J.; Kim, D.H.; Kim, S.G.; Lee, S.Y. Antibacterial Activity of Propolis-Embedded Zeolite Nano-composites for Implant Application.

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
