# Peer review of "Antibacterial Activity of Propolis-Embedded Zeolite Nanocomposites for Implant Application"

_materials, 2021, doi:10.3390/ma14051193_

Round 1
Reviewer 1 Report
Manuscript number: materials-1107065
Title: Antibacterial activity of propolis-embedded zeolite nanocomposites for implant application
This paper describes the potential of propolis-embedded zeolite nanocomposites for dental implant application.
Nanocomposites, based also on biopolymers and ceramic materials, cover a wide range of applications in bioengineering and in regenerative medicine due to their biocompatible properties. However, increasing expectations associated with increasingly longer antibacterial effect, as well as the restrictive requirements to limit the wear of the elements that constitute the implants, make it necessary to still search for ways to improve their usage properties. In this case, it is very important to obtain a biomaterial that will exhibit such properties as excellent bio-functional properties and be able to limit the amount of metal elements (e.g. AgNPs) introduced to an organism. Furthermore, in some branches (e.g. orthopaedics) the material must have an appropriate outer layer structure, what usually limits the disadvantageous of metalosis behaviour. In this work (in Introduction and Results and Discussion parts) this problem has not been described, but the Authors should mention this in the Introduction part. It would be recommended to provide a reference, described these problems, for example: DOI: 10.1002/9781119441632.ch80, 10.1016/j.actbio.2013.06.017.
RESULTS AND DISCUSSION
This section is interesting and good written, but please add more information about antibacterial and cytotoxicity results for the significance of the proposed technology in relation to other, currently used materials in dental implants.
CONCLUSION:
The conclusions must be written in the next (independent) section of the manuscript. Please add more useful information about the novelty and possibility of the application of the proposed technology.
I recommend the paper for publication in Materials but after minor revision, improvement changes are indicated above.
Author Response
Answer to reviewer 1’s comment
Title: Antibacterial activity of propolis-embedded zeolite nanocomposites for implant application
This paper describes the potential of propolis-embedded zeolite nanocomposites for dental implant application.
Nanocomposites, based also on biopolymers and ceramic materials, cover a wide range of applications in bioengineering and in regenerative medicine due to their biocompatible properties. However, increasing expectations associated with increasingly longer antibacterial effect, as well as the restrictive requirements to limit the wear of the elements that constitute the implants, make it necessary to still search for ways to improve their usage properties. In this case, it is very important to obtain a biomaterial that will exhibit such properties as excellent bio-functional properties and be able to limit the amount of metal elements (e.g. AgNPs) introduced to an organism. Furthermore, in some branches (e.g. orthopaedics) the material must have an appropriate outer layer structure, what usually limits the disadvantageous of metalosis behaviour. In this work (in Introduction and Results and Discussion parts) this problem has not been described, but the Authors should mention this in the Introduction part. It would be recommended to provide a reference, described these problems, for example: DOI: 10.1002/9781119441632.ch80, 10.1016/j.actbio.2013.06.017.
Answer) Thanks for your valuable comment. According to your comment, we added some of the issues in the introduction part and also added more citations in the reference section.
In Introduction part
Various nanocomposites including polymer nanoparticles, graphene, zeolites, nanofibers, ceramic materials and metal nanoparticles have been introduced for application in biomedical field due to their unique features [1-5]. Due to their nano-dimensional scales, nanocomposites have biomimetic properties and then covered a wide range of applications in the field of bioengineering and regenerative medicine [1-6]. Especially, nanocomposites or nanomaterials for modification of bioimplant surfaces have been extensively investigated to confer longer antibacterial effect and improved biocompatibilities since infection problems and/or biofilm formation onto implant surfaces induce functional failure and then limit usage as a medical device [7-11]. For example, orthodontic mini-implant coated with chitosan and antibacterial agents effectively inhibited formation of biofilm by Porphyromonas gingivalis [8]. Nanocomposites such as Zno nanorods or nanospheres showed excellent antibacterial activities against Escherichia coli and Staphylococcus aureus [9]. Surface modification with antibacterial peptide such as GL13K provides sustained inhibition of bacterial biofilm formation onto the implant surfaces [10]. Furthermore, nanocomposites and/or nanomaterials for modification of implant surfaces can be considered as an excellent candidate to inhibit metallosis, which causes bone loss around dental implants and systemic cytotoxicity [11-13].
In Discussion part
Additionally, zeolites can be used to prevent liberation of toxic metal ions into human tissues and then improve osteointegration in the bone since metallosis is considered as an important reason for failure of implant and also induces systemic cytotoxicity [12,13,40].
- Shashirekha, G.; Jena, A.; Mohapatra, S. Nanotechnology in Dentistry: Clinical Applications, Benefits, and Hazards. Compend Contin Educ Dent. 2017, 38, e1-e4.
- Anggani, H.S.; Perdana, R.G.; Siregar, E.; Bachtiar, E.W. The effect of coating chitosan on Porphyromonas gingivalis biofilm formation in the surface of orthodontic mini-implant. J. Adv. Pharm. Technol. Res. 2021, 12, 84-88.
- Wang, X.; Fan, H.; Zhang, F.; Zhao, S.; Liu, Y.; Xu, Y.; Wu, R.; Li, D.; Yang, Y.; Liao, L.; Zhu, H.; Wang, X. Antibacterial properties of bilayer biomimetic nano-ZnO for dental implants. ACS Biomater. Sci. Eng. 2020, 6, 1880-1886.
- Holmberg, K.V.; Abdolhosseini, M.; Li, Y.; Chen, X.; Gorr, S.U.; Aparicio, C. Bio-inspired stable antimicrobial peptide coatings for dental applications. Acta Biomater. 2013, 9, 8224-8231.
- Kyzioł, K.; Kaczmarek, Ł.; Kyzioł, A. Surface Functionalization of Biomaterials. In Handbook of Composites from Renewable Materials; Thakur, V.K., Thakur, M.K., Kessler, M.R. Eds; Wiley: Hoboken, USA, 2017; Volume 4, pp. 457-490.
- Wilson, T.G. Jr. Bone loss around implants-is it metallosis? J. Periodontol. 2020, in print.
- Bradberry, S.M.; Wilkinson, J.M.; Ferner, R.E. Systemic toxicity related to metal hip prostheses. Clin. Toxicol. (Phila). 2014, 52, 837-847.
- Bedi, R.S.; Beving, D.E.; Zanello, L.P.; Yan, Y. Biocompatibility of corrosion-resistant zeolite coatings for titanium alloy biomedical implants. Acta Biomater. 2009, 5, 3265-3271.
RESULTS AND DISCUSSION
This section is interesting and good written, but please add more information about antibacterial and cytotoxicity results for the significance of the proposed technology in relation to other, currently used materials in dental implants.
Answer) Thanks for your valuable comment. According to your comment, we added Results and discussion section about antibacterial/cytotoxicity of zeolite-related materials compared to other proposed technology. And references were added.
Additionally, zeolites can be used to prevent liberation of toxic metal ions into human tissues and then improve osteointegration in the bone since metallosis is considered as an important reason for failure of implant and also induces systemic cytotoxicity [12,13,40].
Propolis is proposed as an antibacterial material for dental and oral healthcare because it is a non-toxic material for human uses [41]. Propolis-based mouthwashes have a reasonable antibacterial activity [42,43], However, propolis has limitations in eradication of biofilms because its antibacterial efficacy and toxicity against normal cells are dependent on the metal ions found in propolis [44]. The propolis solution has an appropriate bactericidal effect against S. mutans and C. albicans but it has practically negligible efficacy in eradication of denture biofilms [45]. Afrasiabi et al., reported that propolis nanoparticles provide synergistic effect on the antimicrobial photodynamic therapy against Streptococcus mutans [46]. Cellulose membrane incorporated with propolis-containing self-microemulsifying formulation has sustained release properties for up to 7 days and promotes would healing activity with excellent antibacterial activity [47]. Chitosan vanish containing propolis has sustained release properties over one week and shows improved bactericidal effect compared to CHX [48].
- Bedi, R.S.; Beving, D.E.; Zanello, L.P.; Yan, Y. Biocompatibility of corrosion-resistant zeolite coatings for titanium alloy biomedical implants. Acta Biomater. 2009, 5, 3265-3271.
- Khurshid, Z.; Naseem, M.; Zafar, M.S.; Najeeb, S.; Zohaib, S. Propolis: A natural biomaterial for dental and oral healthcare. J. Dent. Res. Dent. Clin. Dent. Prospects. 2017, 11, 265-274.
- Halboub, E.; Al-Maweri, S.A.; Al-Wesabi, M.; Al-Kamel, A.; Shamala, A.; Al-Sharani, A.; Koppolu, P. Efficacy of propolis-based mouthwashes on dental plaque and gingival inflammation: a systematic review. BMC Oral Health. 2020, 20, 198.
- Santiago, K.B.; Piana, G.M.; Conti, B.J.; Cardoso, E.O.; Murbach Teles Andrade, B.F.; Zanutto, M.R.; Mores Rall, V.L.; Fernandes, A. Jr.; Sforcin, J.M. Microbiological control and antibacterial action of a propolis-containing mouthwash and control of dental plaque in humans. Nat. Prod. Res. 2018, 32, 1441-1445.
- Ambi, A.; Bryan, J.; Borbon, K.; Centeno, D.; Liu, T.; Chen, T.P.; Cattabiani, T.; Traba, C. Are Russian propolis ethanol extracts the future for the prevention of medical and biomedical implant contaminations? Phytomedicine. 2017, 30, 50-58.
- de Souza RF, Silva-Lovato CH, de Arruda CN, Regis RR, Zanini AP, Longo DL, Peracini A, de Andrade IM, Watanabe E, Paranhos HF. Efficacy of a propolis solution for cleaning complete dentures. Am J Dent. 2019 Dec;32(6):306-310.
- Afrasiabi, S.; Pourhajibagher, M.; Chiniforush, N.; Bahador, A. Propolis nanoparticle enhances the potency of antimicrobial photodynamic therapy against Streptococcus mutans in a synergistic manner. Sci. Rep. 2020, 10, 15560.
- Marquele-Oliveira, F.; da Silva Barud, H.; Torres, E.C.; Machado, R.T.A.; Caetano, G.F.; Leite, M.N.; Frade, M.A.C.; Ribeiro, S.J.L.; Berretta, A.A. Development, characterization and pre-clinical trials of an innovative wound healing dressing based on propolis (EPP-AF)-containing self-microemulsifying formulation incorporated in biocellulose membranes. Int. J. Biol. Macromol. 2019, 136, 570-578.
- Franca, J.R.; De Luca, M.P.; Ribeiro, T.G.; Castilho, R.O.; Moreira, A.N.; Santos, V.R.; Faraco, A.A. Propolis—based chitosan varnish: drug delivery, controlled release and antimicrobial activity against oral pathogen bacteria. BMC Complement. Altern. Med. 2014, 14, 478.
CONCLUSION:
The conclusions must be written in the next (independent) section of the manuscript. Please add more useful information about the novelty and possibility of the application of the proposed technology.
Answer) Thanks for your valuable comment. According to your comment, we separate “Conclusion” section in the manuscript. Furthermore, we add more discussion for the novelty and possibility of the application
- Conclusion
This is the first report on the fabrication of propolis-embedded zeolite nanocomposites and their application in dental implant for antibacterial strategy. Chemical properties of propolis were maintained during fabrication process of propolis-embedded zeolite nanocomposites. Propolis was continuously released from propolis-embedded zeolite nanocomposites over one month and PLA/PCl pellets containing propolis-embedded zeolite nanocomposites showed longer sustained release behavior compared to propolis-embedded zeolite nanocomposites. Propolis-embedded zeolite nanocomposite powder showed similar antibacterial activity against C. albicans in agar plate and formed inhibition zone as well as CHX powder. Eluted propolis solution from PLA/PCl pellets also maintained antibacterial activity as well as CHX solution. Eluted propolis solution from PLA/PCl pellets showed meaningful antibacterial efficacy against C. albicans, S. mutans and S. sobrinus. Dental implants fabricated from PLA/PCl polymer with propolis-embedded zeolite nanocomposites also have antibacterial efficacy and negligible intrinsic toxicity against normal cells. Propolis-embedded zeolite nanocomposites can be applied not only as a dental material but also as an orthopedic material due to their excellent antibacterial activity andbiocompatibility.
I recommend the paper for publication in Materials but after minor revision, improvement changes are indicated above.
Answer) Thanks for your valuable comment. We revised the manuscript carefully according to your comment.

Reviewer 2 Report
The manuscript by Prof. Lee et al. describes the formation of hybrid materials containing zeolites, PLA/PCI polymers, and propolis, the latter as an antibacterial agent, and proposes their use as in dental implants. The work is well structured and the experiments are in accordance with what is described in the objectives. Although I recommend its publication in the journal Materials, there are some considerations and corrections that the authors should make before publication:
1) The authors mention "Scheme 1" in the text but there is no such scheme, they should insert it.
2) Figure 1 should include the ATR-FTIR of the zeolite without propolis for comparison, and include a discussion. Also, increase the absorbance of the free propolis sample for better comparison between samples.
3) The authors should justify why in section 3.4 they use E. coli in dental implants, when C. albicans, S. mutans, are more common in oral infections and have been used in previous sections. And if applicable, include the studies with these strains for a better comparison with the results of the previous sections.
4) In the "Discussion" section, things mentioned in the introduction are repeated again and the results obtained are not really discussed. This section should be improved.
If the authors improve the remarks I will be happy to recommend its publication.
Author Response
Answer to reviewer 2’s comment
The manuscript by Prof. Lee et al. describes the formation of hybrid materials containing zeolites, PLA/PCI polymers, and propolis, the latter as an antibacterial agent, and proposes their use as in dental implants. The work is well structured and the experiments are in accordance with what is described in the objectives. Although I recommend its publication in the journal Materials, there are some considerations and corrections that the authors should make before publication:
1) The authors mention "Scheme 1" in the text but there is no such scheme, they should insert it.
Answer) Thanks for your valuable comment. According to your comment, we insert scheme 1. It is our mistake. Thanks again.
Scheme 1. Fabrication of propolis-embedded zeolite nanocomposites.
2) Figure 1 should include the ATR-FTIR of the zeolite without propolis for comparison, and include a discussion. Also, increase the absorbance of the free propolis sample for better comparison between samples.
Answer) Thanks for your valuable comment. According to your comment, we revised the Figure.
Figure 1. ATR-FTIR spectra of propolis-embedded zeolite nanocomposites (a), free propolis (b) and zeolite (c).
3) The authors should justify why in section 3.4 they use E. coli in dental implants, when C. albicans, S. mutans, are more common in oral infections and have been used in previous sections. And if applicable, include the studies with these strains for a better comparison with the results of the previous sections.
Answer) Thanks for your valuable comment. Practically, we choose E. coli (Escherichia coli (E. coli ATCC® 25922TM) to test antibacterial activity of implant for comparison because this stain is susceptible to all the antibiotics and then ease of comparison for antibacterial test.
In Experimental section
Escherichia coli (E. coli ATCC® 25922TM) was obtained from American type culture collection (ATCC, Manassas, USA).
In Results section
Figure 6B shows that the antibacterial activity of real dental implant fabricated in Figure 6A(b) using E. coli.
In Discussion section
Antibacterial activity of real dental implant was evaluated with E. coli ATCC 25922 strain because this stain is susceptible to all the antibiotics and then ease of comparison for antibacterial test [49].
- Gulías, Ò.; McKenzie, G.; Bayó, M.; Agut, M.; Nonell, S. Effective photodynamic inactivation of 26 Escherichia coli strains with different antibiotic susceptibility profiles: a Planktonic and biofilm study. Antibiotics (Basel). 2020, 9, 98.
4) In the "Discussion" section, things mentioned in the introduction are repeated again and the results obtained are not really discussed. This section should be improved. If the authors improve the remarks I will be happy to recommend its publication.
Answer) Thanks for your valuable comment. According to your comment, we fully revised the manuscript in the discussion section.
- Discussion
Zeolite nanoparticles or nanocomposites have investigated by various investigators for application in medical uses [6, 14-17]. Since they have no cytotoxicity against normal cells and tissues, they have been considered as one of the ideal candidate for drug delivery vehicles [14-16,36]. For example, Cao et al., reported that tandem post-synthetic modification of ZIF containing 5-fluorouracil has potential of high drug loading, stimuli-sensitive drug release in tumor microenvironment and synergistic anticancer effect against esophageal squamous cell carcinoma [37]. Chen et al., reported that ammonium methylbenzene blue-incorporated ZIF derivatives have superior antibacterial capacity against E. coli, S. aureus and methicillin-resistant S. aureus [38]. Espcially, ZIF-8 incorporating ceftazidime suggested as a suitable platform for long-term antimicrobial therapy due to its sustained release properties [39]. Furthermore, ZIF-8 incorporating ceftazidime has compatibility against lung epithelial cell lines and then these can be useful platform for targeting of pulmonary and/or intracellular infections since they can be delivered intracellularly against infected normal cells [39]. Additionally, zeolites can be used to prevent liberation of toxic metal ions into human tissues and then improve osteointegration in the bone since metallosis is considered as an important reason for failure of implant and also induces systemic cytotoxicity [12,13,40].
Propolis is proposed as an antibacterial material for dental and oral healthcare because it is a non-toxic material for human uses [41]. Propolis-based mouthwashes have a reasonable antibacterial activity [42,43], However, propolis has limitations in eradication of biofilms because its antibacterial efficacy and toxicity against normal cells are dependent on the metal ions found in propolis [44]. The propolis solution has an appropriate bactericidal effect against S. mutans and C. albicans but it has practically negligible efficacy in eradication of denture biofilms [45]. Afrasiabi et al., reported that propolis nanoparticles provide synergistic effect on the antimicrobial photodynamic therapy against Streptococcus mutans [46]. Cellulose membrane incorporated with propolis-containing self-microemulsifying formulation has sustained release properties for up to 7 days and promotes would healing activity with excellent antibacterial activity [47]. Chitosan vanish containing propolis has sustained release properties over one week and shows improved bactericidal effect compared to CHX [48]. In our study, propolis-embedded zeolite nanocomposites showed sustained release properties over one month as shown in Figure 3(b). It seems that drug release capacity of our nanocomposites was almost similar to those of other results [47]. Marquele-Oliveira et al., reported that drug release from biocellulose membrane was continued over one week [47]. Furthermore, Franca et al., also reported that propolis was continuously released from chitosan vanish over two months [48]. Our PLA/PCL pellets incorporated with propolis-embedded zeolite nanocomposites also showed extended drug release behavior over one month, indicating that propolis-embedded zeolite nanocomposites and their polymer pellets are suitable device for long-term inhibition of dental/oral pathogens in implant. Furthermore, PLA/PCL pellets complexed with propolis-embedded zeolite nanocomposites showed longer sustained release behavior compared to zeolite nanocomposites. Our dental implant platforms also have long-term treatment for antibacterial strategy. Antibacterial activity of real dental implant was evaluated with E. coli ATCC 25922 strain because this stain is susceptible to all the antibiotics and then ease of comparison for antibacterial test [49]. We proved that dental implant fabricated from PLA/PCL polymer and propolis-embedded zeolite nanocomposites has evident antibacterial efficacy against E. coli (Figure 6B). Furthermore, these platforms have no significant cell cytotoxicity against HaKat cells (Figure 7). Our results were quite similar to the results of other investigators. Since bacterial infection and/or biofilm formation onto the dental implant is closely associated with dental disease and failure of dental implant, long-term treatment strategy using sustained release implant modules are considered as an ideal candidate for success and maintenance of dental implant.
Propolis and/or its extracts have antitumor, antibacterial and anti-inflammatory effects in biological systems [50-52]. Especially, mouthwash application of propolis is known to suppress dysphagia or mucositis efficiently in clinical trials [42,43]. Its long-term antibacterial and anti-inflammatory effects may helpful to settlement of dental implant. Furthermore, it was reported that propolis nanoparticles have high efficacy to suppress S. mutans-derived biofilm formation with negligible cytotoxicity against HGF-1 human gingival fibroblast cells [46]. Our results indicated that dental implant fabricated from PLA/PCL polymer and propolis-embedded zeolite nanocomposites maintained antibacterial activity during pellet fabrication process and, furthermore, has no significant cytotoxicity against normal cells (Figure 6 and 7).

Round 2
Reviewer 2 Report
Thanks to the authors for following my advice.